# Increased EBV DNAemia after Anti-SARS-CoV-2 Vaccination in Solid Organ Transplants

**DOI:** 10.3390/vaccines10070992

**Published:** 2022-06-22

**Authors:** Joanna Musialik, Aureliusz Kolonko, Andrzej Więcek

**Affiliations:** Department of Nephrology, Transplantation and Internal Medicine, Medical University of Silesia in Katowice, Francuska Str. 20-24, 40-027 Katowice, Poland; uryniusz@wp.pl (A.K.); awiecek@sum.edu.pl (A.W.)

**Keywords:** Epstein–Barr virus, solid organ transplant, COVID-19, vaccine

## Abstract

The reactivation of latent viruses during SARS-CoV-2 infection is well recognized, and coinfection with Epstein–Barr virus (EBV) has been associated with severe clinical cases of COVID-19 infection. In transplant patients, EBV infection presents a significant challenge. Assessing the potential impact of SARS-CoV-2 vaccinations on EBV infections in stable kidney and liver transplant recipients was the objective of our study. Ten solid-organ-transplant (SOT) patients (eight kidney and two liver) vaccinated with standard doses of mRNA COVID-19 vaccines were included. EBV DNA viral load measurements were conducted prior to the vaccination and during a follow-up period (at the first month and after six months) after the second vaccine dose. After the second dose, a significant increase in median viremia was observed (*p* < 0.01) in 9 patients, and in one patient, the reactivation of EBV infection was found. Six months later, the median viremia decreased significantly (*p* < 0.05). The EBV viral load should be closely monitored as it could lead to the earlier diagnosis and treatment of EBV-related complications. Despite experiencing a decrease in the viral load six months post-vaccination, some patients still had a viral load over the baseline, which increased the risk of potential complications.

## 1. Introduction

The ongoing COVID-19 pandemic presents a serious threat to the global population due to its high contagiousness and mortality. Among the most affected have been the elderly and those with comorbidities such as chronic heart, lung, and kidney insufficiencies [1,2]. Solid-organ-transplant (SOT) recipients are a high-risk population due to both frequent comorbidities and their compromised immunity resulting from the immunosuppressive regimen required for successful organ transplantation. Recently, the deleterious effect of the pandemic has been reduced by the development of effective vaccination strategies. However, as millions of people worldwide have been receiving anti-SARS-CoV-2 vaccines, the information concerning their adverse effects has been increasing as well [3,4].

Epstein–Barr virus (EBV) belongs to a human herpes virus family. It is typically latent in more than 90% of the global population [5], but in SOT recipients, post-transplant EBV infection or EBV reactivation can cause serious complications, including the worsening of the graft function and post-transplant lymphoproliferative disease (PTLD) [6,7]. The main risk factor of EBV infection is associated with the net strength of the immunosuppressive regimen, the reduction of which is also the key strategy for treatment [8]. According to the recent literature, vaccination for SARS-CoV-2 in the general population has resulted in the reactivation of other herpes viruses, including cytomegalovirus and varicella zoster virus [9,10,11], in the general population. Moreover, rapidly growing EBV-positive lymphoma were reported seven days after the first dose of a COVID-19 vaccine [12]. As our early clinical observations have suggested substantial increases in EBV viremia in transplant patients, we investigated the effect of anti-SARS-CoV-2 mRNA vaccinations on EBV viremia in stable kidney and liver transplant recipients.

## 2. Patients and Methods

The present analysis was performed in all transplant patients who regularly attended our out-patient transplant clinic and for whom the appropriate EBV DNA results were available. EBV DNA viral load measurements were performed using plasma quantitative polymerase chain reaction (PCR) (Artus EBV PCR Kits RUO, Quiagen, Germany) with a sensitive detection threshold of 157 copies/mL. We analyzed the last two consecutive EBV titers prior to vaccination (within 2 months and within 12 months prior to vaccination), and their median value was set as a baseline value in this study. During the follow-up period, we analyzed the first and second EBV viremia measurements performed after the second dose of vaccine (at the first month and after six months). At the same time, we assessed the function of the transplanted organ using serum creatinine level and estimated glomerular filtration rate (eGFR) in kidney patients as well as aminotransferases, γ-glutamyltranspeptidase (GGT), bilirubin, and international normalized ratio (INR) in liver transplant recipients. The absolute number and percentage of lymphocytes and blood level of immunosuppression drugs were also analyzed. All patients were vaccinated with the Pfizer BioNTech COVID-19 vaccine, with the exception of one who received the Moderna mRNA01273 COVID-19 vaccine. All patients received standard doses of vaccines at time intervals consistent with the manufactures’ recommendations (in our group, the second dose was given at least 4 weeks after the first vaccination).

Finally, ten SOT patients (8 kidney and 2 liver, 8F, median age 40 (IQR 36–54) years, median time after transplantation 68 (49–111 months) were included in the analysis. 

Statistical analyses were performed using STATISTICA 13.0 PL for Windows software package (StatSoft Poland, Cracow, Poland). Data were presented as medians with interquartile ranges (IQRs). The differences in EBV viremia and kidney or liver graft function parameters were calculated using the Wilcoxon test for paired samples. 

## 3. Results

In the single liver transplant patient who had previously been EBV-negative for an extended period, the reactivation of EBV infection with viremia over 10^3^ copies/mL (1273 copies/mL) occurred after a second dose of vaccine, and 6 months later, the viral load continued to exceed 10^3^ copies/mL. Concomitantly, the activity of alanine and aspartate aminotransferases as well as γ-glutamyltranspeptidase increased (87 vs. 112 IU/mL; 54 vs. 69 IU/mL; and 136 vs. 207 IU/mL, respectively), whereas the bilirubin levels and the INR values were unchanged. 

In the other 9 patients, after the second dose of vaccine, a significant increase in median viremia was observed (7062 copies/mL (IQR 2321–79,285) vs. 54,451 copies/mL (IQR 19,655–225,990); *p* < 0.01) (Figure 1). The individual results of the EBV viremia changes are presented in Figure 2. During the six-month post-vaccination observation period, the kidney and liver graft functions were stable in all patients, as were the levels of immunosuppressive drugs. The median viremia of 20,200 copies/mL (IQR 565–34,070) was observed 6 months after the second dose (*p* < 0.05 vs. the results at first month post-vaccination and without significance vs. the initial results).

## 4. Discussion

We presented a series of 10 transplant recipients with noticeable EBV viremia exacerbation or reactivation within a short period after two doses of COVID-19 mRNA vaccines. To date, the connection between COVID-19 infection and EBV infection has only been postulated. In the general population, Paolucci et al. investigated the frequency of different opportunistic viral infections in COVID-19 patients and found a high incidence of EBV viremia [13]. Similarly, in critically ill patients with COVID-19, EBV was the most common herpes virus co-infection [14]. EBV/SARS-CoV-2 coinfection patients have been characterized by a higher risk of fever symptom and higher ALT activity, as compared to subjects with only SARS-CoV-2 infections [15]. Finally, immunocompetent patients with “long” COVID-19 symptoms, as well as potentially serious complications such as myocarditis, cardiomyopathy, acute liver injury, and hemolytic anemia, were described in [16].

A few reports have been published concerning the reactivation of different viral infections, including varicella-zoster and herpes simplex viruses after vaccination for COVID-19 [17,18,19]. As it raised questions concerning the vaccine’s safety, Brosh-Nissimov et al. investigated the oropharyngeal shedding of herpes viruses before and after BNT162b2 mRNA vaccination [20]. Interestingly, they did not find evidence for increased reactivation of herpes viruses within one week after vaccine. However, the authors noted that herpes viruses, including EBV, may reactivate in other unexplored sites, such as trigeminal ganglion, facial nerves, and skin [20] Nonetheless, the mechanism of the reactivation of long-time latent or stable viral infection is not yet fully understood, but this phenomenon could apply to other infections and other vaccinations. In our center, we have observed a young, healthy patient with chronic hepatitis B infection and HBV viral load, who was stable after many years at approximately 1500 copies/mL but abruptly increased to more than 100,000 copies/mL after vaccination against yellow fever. The viral load returned to baseline for three months.

During natural COVID-19 infection, both cytokine storm and lymphopenia may occur, which could trigger subsequent viral reactivation [17]. It may be also potentially associated with poor immune surveillance of the previous infection while building resistance to a novel infection. Nevertheless, the vast majority of patients receiving COVID-19 mRNA vaccinations have not suffered from lymphopenia or cytokine activation. Importantly, it was shown that EBV reactivation and both B cell and epithelial cell differentiation were closely associated [16]. Another study suggested the potential role of oxidative stress and psychological stressors on the immune system. Hence, EBV could function as an opportunistic virus and could take advantage of host co-infection with CMV, syphilis, and human papillomavirus to reactivate [16]. Therefore, it is possible that vaccine-stimulated immunomodulation increased the possibility of viral reactivation [17]. 

Notably, in the stable SOT cohort, EBV viremia was detected in 4.1% patients, with EBV D + R-(donor-recipient) status as a main risk factor of PTLD [21]. Moreover, all except one patient in our present report had EBV viremia prior to COVID-19 vaccination. We hypothesized that the maintenance immunosuppression in our stable SOT recipients predisposed them to such a significant increase in EBV viremia post-vaccination. It should be added that none of the observed patients presented symptoms of EBV disease after vaccination and so far did not show any signs of PTLD.

This observation is important for several reasons. First, the relationship between the EBV viral load and the risk of serious complications, especially PTLD, has been previously established [8,22]. The identified threshold of the EBV viral load associated with a significantly higher risk of PTLD was assessed at 4000 copies/mL [23], especially in high-risk subjects and patients treated with mycophenolic acid [24]. In our group, the pre-vaccination viral load exceeded 4000 copies/mL in 6 patients, however, none of these patients had signs of PTLD despite careful monitoring. Nevertheless, the risk of a sustained, significant increase in EBV viral load in this group of patients is much higher, which significantly increases the risk of developing PTLD. These patients require further systematic viral load monitoring and much closer clinical supervision. Secondly, a specific bundle of symptoms, including fatigue, brain fog, and rashes (“long” COVID-19 symptoms), was recently described in some patients after the resolution of COVID-19 infection [25]. Other persistent symptoms could involve chest and joint pains, palpitations, myalgia, cough, and gastrointestinal and cardiac issues. Proposed pathophysiology included direct tissue damage and inflammation as a result of viral persistence, immune dysregulation, and autoimmunity) [25]. However, current findings have suggested that many long COVID-19 symptoms may not be the result of the SARS-CoV-2 virus but may be a consequence of EBV reactivation secondary to COVID-19 infection [26]. 

The problem may be more universal and affect many other latent viruses, but up until now, very few cases of reactivation of herpes virus infection after the vaccination for hepatitis A, influenza, and rabies [27] have been published. It is possible that only the global universality of the vaccination against COVID-19 allows us to draw attention to this potential problem. The major drawback of our study was its small sample size and that we were only able to analyse those SOT recipients in whom the EBV viral load has previously been monitored within the last year prior to being vaccinated for COVID-19. However, our pioneering study identified a significant increase in EBV viremia among SOT patients that occurred after two doses of COVID mRNA vaccine.

## 5. Conclusions

In summary, considering the ongoing necessity for extensive vaccination in immunocompromised transplant populations and the potentially dangerous side effects, such as post-vaccine EBV activation, close monitoring of EBV viral load should be advocated as it could lead to earlier diagnosis and treatment of EBV-relates complications. This is particularly critical due to the higher viral load (greater than baseline) found in some of our patients 6 months post-vaccination, which thus increased their potential for complications. Furthermore, a prospective study investigating EBV viral load fluctuations after COVID-19 vaccination, as well as its potentially harmful consequences, is warranted.

## Figures and Tables

**Figure 1 vaccines-10-00992-f001:**
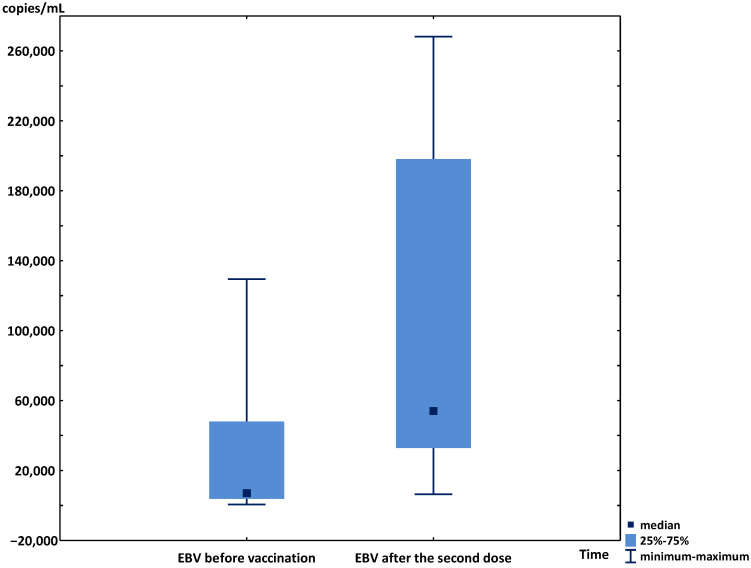
Median, minimum, maximum, and interquartile ranges of EBV viremia before the vaccination and after the second dose of SARS-CoV-2 vaccine in transplant patients.

**Figure 2 vaccines-10-00992-f002:**
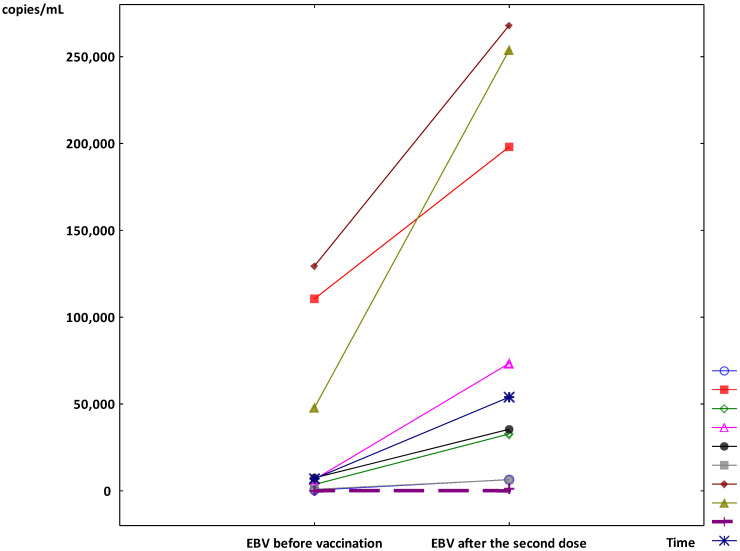
The individual results of the Epstein–Barr virus (EBV) viremia during the follow-up period (The individual symbols mark the next patients and the dashed line marks the results of the patient who reactivated EBV infection).

## Data Availability

The data presented in this study are available are available within the article.

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
