# Peer review of "Increased EBV DNAemia after Anti-SARS-CoV-2 Vaccination in Solid Organ Transplants"

_vaccines, 2022, doi:10.3390/vaccines10070992_

Round 1
Reviewer 1 Report
The authors found after the second dose of COVID-19 vaccination, a significant increase in median viremia was observed, and the reactivation of EBV infection was found. This is not a new report, as the authors have cited, EBV rebound after COVID-19 vaccination was already observed in transplant patients. The following suggestion may improve the quality of the paper.
If the EBV rebound after vaccination is this COVID-19 vaccine-specific? How about influenza vaccination? EBV infected B cells and can be a latent stage, However, when the B cells were activated, the latent stage of the virus would be active replication. The EBV rebound after the COVID-19 vaccination may only reflect the immune activation and some latent stage EBV was reactivated in B cells. It means any infection or vaccination would also induce EBV rebound.
Is the COVID-19 vaccination-specific induced EBV? I think probably not. Like the point above, you may detect many viruses rebound if these viruses can be in the latent stages in PBMCs. Such as HIV virus in HIV patients would be rebound after COVID-19 vaccination.
If growing EBV-positive is transplanted patients specifically? If rapidly growing EBV-positive would be observed in healthy control? My opinion is still not transplanted patients specifically. It only reflects B cell activation. It would happen in any individual that EBV positive.
Author Response
Thank you for your comments and suggestions.
Ad 1. – Indeed, one paper has emerged so far pointing to a potential association with rapidly progressing PTLD after vaccination in two heart transplant patients. This work was quoted in our article.
Ad 2. Theoretically, EBV reactivation may occur after each vaccination, as well as spontaneous reactivations or exacerbations related to, for example, the virus life cycle. However, only in permanently immunocompromised patients this can lead to a persistent increase in viral load and an increased risk of PTLD. In healthy subjects, this is not clinically significant in the vast majority of cases. Certainly, however, as in the case of a patient with HBV, there is a temporary over-modulation of the immune system towards a new infection after vaccination of any kind. However, we have not found only one report on the reactivation of latent viruses after other vaccinations but not in in transplant patients - we added this observation to our paper. Observations of the reactivation of latent viruses after anti-SARS vaccination, not only in transplanted patients, may be associated with the mass vaccination or the uniqueness of the vaccine developed in such a short time and thus arising a lot of emotions. Thus, indeed, it seems possible that potentially any vaccination also induces an EBV rebound effect.
3. We also think the virus is not specific. However, in the case of, for example, HIV-infected patients who are constantly receiving antiretroviral therapy, I believe that any rebound of HIV has not been established, but we have not found relevant publications.
4. Certainly, all people infected with EBV present transient increase in viral load for various reasons, including other infections or due to chronic stress (Sausen DG, Bhutta MS, Gallo ES, Dahari H, Borenstein R. Stress-Induced Epstein-Barr Virus Reactivation. Biomolecules. 2021; 11 (9): 1380. However, when the immune system is properly functioning, in the great majority of cases it limits the further increase in viral load. In the case of transplant patients, especially those suffering from combined immunosuppression, the situation is completely different.
Reviewer 2 Report
I am grateful for the opportunity to review the manuscript by Musialik et al. The manuscript is well written and brings a very interesting theme to the current moment of the COVID-19 pandemic. The methodology, results and discussion are adequate. I suggest that authors should include the vaccine that was included in the study (mRNA).
Author Response
Thank you for the warm welcome and high appreciation of our work.
Reviewer 3 Report
The global pandemic of COVID-19 is causing a public health crisis, especially in high-risk populations including solid organ transplant recipients. It is not only due to their compromised immunity but also to frequent comorbidities. Substantial increases in EBV viremia had been found in transplant patients, which can cause serious complications.
To assess the effect of COVID-19 mRNA vaccinations on EBV viremia in stable kidney and liver transplant recipients, the authors of this manuscript compared the EBV DNA copy number of ten transplant patients after and before COVID-19 vaccination. They found a significant increase in median viremia in 9 patients, as well as the reactivation of latent EBV in one patient. This is an interesting topic and an important issue for assessing the side effects of the COVID-19 vaccination on SOT recipients. Although there are no conclusions that can be drawn from the study, due to a small number of cases and limited data, it does provide some information on this issue. It is helpful for understanding the adverse effect of the COVID-19 vaccine in transplant patients. Furthermore, I have some concerns that need to be addressed.
1. In line 67, the exact time should be presented in the text.
2. At least three patients had active EBV infection prior to vaccination, and their copy number was higher than in other patients post-vaccination. The authors should discuss the significance of the two situations in the manuscript.
3. The title of this report is inaccurate and slightly exaggerated. Please revise it accordingly.
4. Figure 1 does not clearly depict the comparison of EBV viremia pre- and post-vaccination. Individual plots with mean and standard deviation may provide a more accurate and clear representation of the data.
Author Response
Thank you for your comments and suggestions.
All your suggestions we we have included in the text
Round 2
Reviewer 1 Report
No further questions.